# Robust Attribution Regularization

**Jiefeng Chen** [*1]    **Xi Wu** [*2]    **Vaibhav Rastogi** [†2]    **Yingyu Liang** [1]    **Somesh Jha** [1,3]

[1] University of Wisconsin-Madison    [2] Google    [3] XaiPient

## Abstract

An emerging problem in trustworthy machine learning is to train models that produce robust interpretations for their predictions. We take a step towards solving this problem through the lens of axiomatic attribution of neural networks. Our theory is grounded in the recent work, *Integrated Gradients* (IG) [STY17], in *axiomatically attributing* a neural network's *output change* to *its input change*. We propose training objectives in classic robust optimization models to achieve robust IG attributions. Our objectives give principled generalizations of previous objectives designed for robust predictions, and they naturally degenerate to classic soft-margin training for one-layer neural networks. We also generalize previous theory and prove that the objectives for different robust optimization models are closely related. Experiments demonstrate the effectiveness of our method, and also point to intriguing problems which hint at the need for better optimization techniques or better neural network architectures for robust attribution training.

## 1   Introduction

Trustworthy machine learning has received considerable attention in recent years. An emerging problem to tackle in this domain is to train models that produce reliable interpretations for their predictions. For example, a pathology prediction model may predict certain images as containing malignant tumor. Then one would hope that under visually indistinguishable perturbations of an image, similar sections of the image, instead of entirely different ones, can account for the prediction. However, as Ghorbani, Abid, and Zou [GAZ17] convincingly demonstrated, for existing models, one can generate minimal perturbations that substantially change model interpretations, *while keeping their predictions intact*. Unfortunately, while the *robust prediction* problem of machine learning models is well known and has been extensively studied in recent years (for example, [MMS+17a, SND18, WK18], and also the tutorial by Madry and Kolter [KM18]), there has only been limited progress on the problem of *robust interpretations*.

In this paper we take a step towards solving this problem by viewing it through the lens of axiomatic attribution of neural networks, and propose Robust Attribution Regularization. Our theory is grounded in the recent work, *Integrated Gradients* (IG) [STY17], in *axiomatically attributing* a neural network's *output change* to *its input change*. Specifically, given a model $f$, two input vectors $\boldsymbol{x}, \boldsymbol{x}'$, and an input coordinate $i$, $\mathrm{IG}_i^f(\boldsymbol{x}, \boldsymbol{x}')$ defines a path integration (parameterized by a curve from $\boldsymbol{x}$ to $\boldsymbol{x}'$) that assigns a number to the $i$-th input as its "contribution" to the change of the model's output from $f(\boldsymbol{x})$ to $f(\boldsymbol{x}')$. IG enjoys several natural theoretical properties (such as the Axiom of Completeness[3]) that other related methods violate.

---

[*]Equal contribution.

[†]Work done while at UW-Madison.

Due to lack of space and for completeness, we put some definitions (such as coupling) to Section B.1. Code for this paper is publicly available at the following repository: https://github.com/jfc43/robust-attribution-regularization

[3]Axiom of Completeness says that summing up attributions of all components should give $f(\boldsymbol{x}') - f(\boldsymbol{x})$.

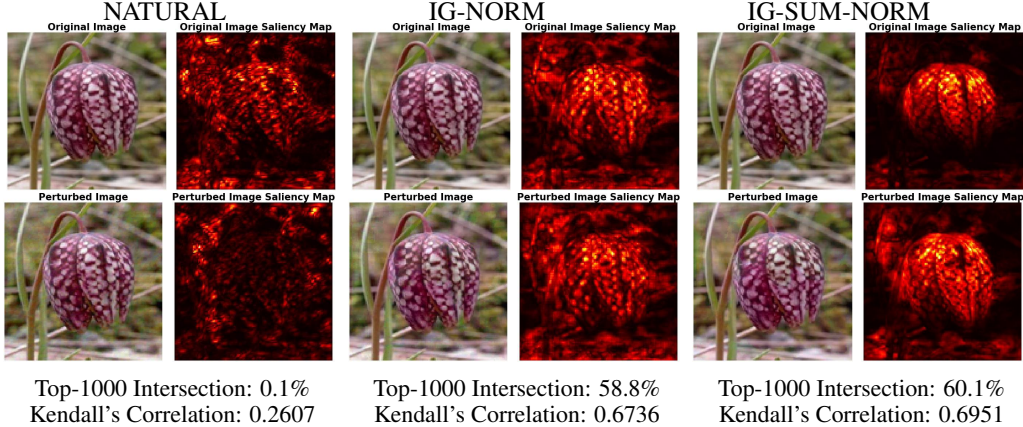

NATURAL

IG-NORM

IG-SUM-NORM

Top-1000 Intersection: 0.1%
Kendall's Correlation: 0.2607

Top-1000 Intersection: 58.8%
Kendall's Correlation: 0.6736

Top-1000 Intersection: 60.1%
Kendall's Correlation: 0.6951

Figure 1: **Attribution robustness comparing different models.** Top-1000 Intersection and Kendall's Correlation are rank correlations between original and perturbed saliency maps. NATURAL is the naturally trained model, IG-NORM and IG-SUM-NORM are models trained using our robust attribution method. We use attribution attacks described in [GAZ17] to perturb the attributions while keeping predictions intact. For all images, the models give *correct* prediction – Windflower. However, the saliency maps (also called feature importance maps), computed via IG, show that attributions of the naturally trained model are very fragile, either visually or quantitatively as measured by correlation analyses, while models trained using our method are much more robust in their attributions.

We briefly overview our approach. Given a loss function $\ell$ and a data generating distribution $P$, our Robust Attribution Regularization objective contains two parts: **(1)** Achieving a small loss over the distribution $P$, and **(2)** The IG attributions of the loss $\ell$ over $P$ are "close" to the IG attributions over $Q$, if distributions $P$ and $Q$ are close to each other. We can naturally encode these two goals in two classic robust optimization models: **(1)** In the *uncertainty set model* [BTEGN09] where we treat sample points as "nominal" points, and assume that true sample points are from certain vicinity around them, which gives:

$$\underset{\theta}{\text{minimize}} \quad \underset{(\boldsymbol{x},y)\sim P}{\mathbb{E}}[\rho(\boldsymbol{x},y;\theta)]$$

$$\text{where } \rho(\boldsymbol{x},y;\theta) = \ell(\boldsymbol{x},y;\theta) + \lambda \max_{\boldsymbol{x}'\in N(\boldsymbol{x},\varepsilon)} s(\text{IG}_{\boldsymbol{h}}^{\ell_y}(\boldsymbol{x},\boldsymbol{x}';r))$$

where $\text{IG}_{\boldsymbol{h}}^{\ell_y}(\cdot)$ is the attribution w.r.t. neurons in an intermediate layer $\boldsymbol{h}$, and $s(\cdot)$ is a size function (e.g., $\|\cdot\|_2$) measuring the size of IG, and **(2)** In the *distributional robustness* model [SND18, MEK15], where closeness between $P$ and $Q$ is measured using metrics such as Wasserstein distance, which gives:

$$\underset{\theta}{\text{minimize}} \quad \mathbb{E}_P[\ell(P;\theta)] + \lambda \sup_{Q;M\in\prod(P,Q)} \left\{ \underset{Z,Z'}{\mathbb{E}}[d_{\text{IG}}(Z,Z')] \text{ s.t. } \underset{Z,Z'}{\mathbb{E}}[c(Z,Z')] \leq \rho \right\},$$

In this formulation, $\prod(P,Q)$ is the set of couplings of $P$ and $Q$, and $M = (Z,Z')$ is one coupling. $c(\cdot,\cdot)$ is a metric, such as $\|\cdot\|_2$, to measure the cost of an adversary perturbing $z$ to $z'$. $\rho$ is an upper bound on the expected perturbation cost, thus constraining $P$ and $Q$ to be "close" with each together. $d_{\text{IG}}$ is a metric to measure the change of attributions from $Z$ to $Z'$, where we want a large $d_{\text{IG}}$-change under a small $c$-change. The supremum is taken over $Q$ and $\prod(P,Q)$.

We provide theoretical characterizations of our objectives. First, we show that they give principled generalizations of previous objectives designed for *robust predictions*. Specifically, under *weak* instantiations of size function $s(\cdot)$, and how we estimate IG computationally, we can leverage axioms satisfied by IG to recover the robust prediction objective of [MMS+17a], the input gradient regularization objective of [RD18], and also the distributional robust prediction objective of [SND18]. These results provide theoretical evidence that robust prediction training can provide some control over robust interpretations. Second, for one-layer neural networks, we prove that instantiating $s(\cdot)$ as 1-norm coincides with the instantiation of $s(\cdot)$ as sum, and further coincides with classic soft-margin

training, which implies that for generalized linear classifiers, soft-margin training will robustify both predictions and interpretations. Finally, we generalize previous theory on distributional robust prediction [SND18] to our objectives, and show that they are closely related.

Through detailed experiments we study the effect of our method in robustifying attributions. On MNIST, Fashion-MNIST, GTSRB and Flower datasets, we report encouraging improvement in attribution robustness. Compared with naturally trained models, we show significantly improved attribution robustness, as well as prediction robustness. Compared with Madry et al.'s model [MMS$^+$17a] trained for robust predictions, we demonstrate *comparable* prediction robustness (*sometimes even better*), while *consistently* improving attribution robustness. We observe that even when our training stops, the attribution regularization term remains much more significant compared to the natural loss term. We discuss this problem and point out that current optimization techniques may not have effectively optimized our objectives. These results hint at the need for better optimization techniques or new neural network architectures that are more amenable to robust attribution training.

The rest of the paper is organized as follows: Section 2 briefly reviews necessary background. Section 3 presents our framework for robustifying attributions, and proves theoretical characterizations. Section 4 presents instantiations of our method and their optimization, and we report experimental results in Section 5. Finally, Section 6 concludes with a discussion on future directions.

## 2 Preliminaries

**Axiomatic attribution and Integrated Gradients** Let $f : \mathbb{R}^d \mapsto \mathbb{R}$ be a real-valued function, and $\boldsymbol{x}$ and $\boldsymbol{x}'$ be two input vectors. Given that function values changes from $f(\boldsymbol{x})$ to $f(\boldsymbol{x}')$, a basic question is: *"How to attribute the function value change to the input variables?"* A recent work by Sundararajan, Taly and Yan [STY17] provides an *axiomatic answer* to this question. Formally, let $r : [0, 1] \mapsto \mathbb{R}^d$ be a curve such that $r(0) = \boldsymbol{x}$, and $r(1) = \boldsymbol{x}'$, Integrated Gradients (IG) for input variable $i$ is defined as the following integral:

$$\text{IG}_i^f(\boldsymbol{x}, \boldsymbol{x}'; r) = \int_0^1 \frac{\partial f(r(t))}{\partial \boldsymbol{x}_i} r_i'(t) dt, \tag{1}$$

which formalizes the contribution of the $i$-th variable as the integration of the $i$-th partial as we move along curve $r$. Let $\text{IG}^f(\boldsymbol{x}, \boldsymbol{x}'; r)$ be the vector where the $i$-th component is $\text{IG}_i^f$, then $\text{IG}^f$ satisfies some natural axioms. For example, the Axiom of Completeness says that summing all coordinates gives the change of function value: $\text{sum}(\text{IG}^f(\boldsymbol{x}, \boldsymbol{x}'; r)) = \sum_{i=1}^d \text{IG}_i^f(\boldsymbol{x}, \boldsymbol{x}'; r) = f(\boldsymbol{x}') - f(\boldsymbol{x})$. We refer readers to the paper [STY17] for other axioms IG satisfies.

**Integrated Gradients for an intermediate layer.** We can generalize the theory of IG to an intermediate layer of neurons. The key insight is to leverage the fact that Integrated Gradients is a *curve integration*. Therefore, given some hidden layer $\boldsymbol{h} = [h_1, \ldots, h_l]$, computed by a function $h(\boldsymbol{x})$ induced by previous layers, one can then naturally view the previous layers as inducing a *curve* $h \circ r$ which moves from $h(\boldsymbol{x})$ to $h(\boldsymbol{x}')$, as we move from $\boldsymbol{x}$ to $\boldsymbol{x}'$ along curve $r$. Viewed this way, we can thus naturally compute IG for $\boldsymbol{h}$ in a way that leverages all layers of the network[4],

**Lemma 1.** *Under curve $r : [0, 1] \mapsto \mathbb{R}^d$ such that $r(0) = \boldsymbol{x}$ and $r(1) = \boldsymbol{x}'$ for moving $\boldsymbol{x}$ to $\boldsymbol{x}'$, and the function induced by layers before $\boldsymbol{h}$, the attribution for $h_i$ for a differentiable $f$ is*

$$\text{IG}_{h_i}^f(\boldsymbol{x}, \boldsymbol{x}') = \sum_{j=1}^d \left\{ \int_0^1 \frac{\partial f(h(r(t)))}{\partial h_i} \frac{\partial h_i(r(t))}{\partial \boldsymbol{x}_j} r_j'(t) dt \right\}. \tag{2}$$

*The corresponding summation approximation is:*

$$\text{IG}_{h_i}^f(\boldsymbol{x}, \boldsymbol{x}') = \frac{1}{m} \sum_{j=1}^d \left\{ \sum_{k=0}^{m-1} \frac{\partial f(h(r(k/m)))}{\partial h_i} \frac{\partial h_i(r(k/m))}{\partial \boldsymbol{x}_j} r_j'(k/m) \right\} \tag{3}$$

## 3 Robust Attribution Regularization

In this section we propose objectives for achieving robust attribution, and study their connections with existing robust training objectives. At a high level, given a loss function $\ell$ and a data generating

distribution $P$, our objectives contain two parts: (1) Achieving a small loss over the data generating distribution $P$, and (2) The IG attributions of the loss $\ell$ over $P$ are "close" to the IG attributions over distribution $Q$, if $P$ and $Q$ are close to each other. We can naturally encode these two goals in existing robust optimization models. Below we do so for two popular models: the *uncertainty set model* and the *distributional robustness* model.

## 3.1 Uncertainty Set Model

In the uncertainty set model, for any sample $(\boldsymbol{x}, y) \sim P$ for a data generating distribution $P$, we think of it as a "nominal" point and assume that the real sample comes from a neighborhood around $\boldsymbol{x}$. In this case, given any intermediate layer $\boldsymbol{h}$, we propose the following objective function:

$$
\begin{aligned}
&\underset{\theta}{\text{minimize}} \quad \underset{(\boldsymbol{x}, y) \sim P}{\mathbb{E}}[\rho(\boldsymbol{x}, y; \theta)] \\
&\text{where } \rho(\boldsymbol{x}, y; \theta) = \ell(\boldsymbol{x}, y; \theta) + \lambda \max_{\boldsymbol{x}' \in N(\boldsymbol{x}, \varepsilon)} s(\text{IG}_{\boldsymbol{h}}^{\ell_y}(\boldsymbol{x}, \boldsymbol{x}'; r))
\end{aligned}
\tag{4}
$$

where $\lambda \geq 0$ is a regularization parameter, $\ell_y$ is the loss function with label $y$ fixed: $\ell_y(\boldsymbol{x}; \theta) = \ell(\boldsymbol{x}, y; \theta)$, $r : [0, 1] \mapsto \mathbb{R}^d$ is a curve parameterization from $\boldsymbol{x}$ to $\boldsymbol{x}'$, and $\text{IG}^{\ell_y}$ is the integrated gradients of $\ell_y$, and therefore gives attribution of changes of $\ell_y$ as we go from $\boldsymbol{x}$ to $\boldsymbol{x}'$. $s(\cdot)$ is a size function that measures the "size" of the attribution.[5]

We now study some particular instantiations of the objective (4). Specifically, we recover existing robust training objectives under *weak* instantiations (such as choosing $s(\cdot)$ as summation function, which is not metric, or use crude approximation of IG), and also derive new instantiations that are natural extensions to existing ones.

**Proposition 1 (Madry et al.'s robust prediction objective).** *If we set $\lambda = 1$, and let $s(\cdot)$ be the* sum *function (sum all components of a vector), then for any curve $r$ and any intermediate layer $\boldsymbol{h}$, (4) is exactly the objective proposed by Madry et al. [MMS$^+$17a] where $\rho(\boldsymbol{x}, y; \theta) = \max_{\boldsymbol{x}' \in N(\boldsymbol{x}, \varepsilon)} \ell(\boldsymbol{x}', y; \theta)$.*

We note that: (1) sum is a weak size function which does not give a metric. (2) As a result, while this robust prediction objective falls within our framework, and regularizes robust attributions, it allows a small regularization term where attributions actually change significantly but they cancel each other in summation. Therefore, the control over robust attributions can be weak.

**Proposition 2 (Input gradient regularization).** *For any $\lambda' > 0$ and $q \geq 1$, if we set $\lambda = \lambda'/\varepsilon^q$, $s(\cdot) = \|\cdot\|_1^q$, and use only the first term of summation approximation (3) to approximate IG, then (4) becomes exactly the input gradient regularization of Drucker and LeCun [DL92], where we have $\rho(\boldsymbol{x}, y; \theta) = \ell(\boldsymbol{x}, y; \theta) + \lambda \|\nabla_{\boldsymbol{x}} \ell(\boldsymbol{x}, y; \theta)\|_q^q$.*

In the above we have considered instantiations of a weak size function (summation function), which recovers Madry et al.'s objective, and of a weak approximation of IG (picking the first term), which recovers input gradient regularization. In the next example, we pick a nontrivial size function, the 1-norm $\|\cdot\|_1$, use the precise IG, but then we use a *trivial intermediate layer*, the output loss $\ell_y$.

**Proposition 3 (Regularizing by attribution of the loss output).** *Let us set $\lambda = 1$, $s(\cdot) = \|\cdot\|_1$, and $\boldsymbol{h} = \ell_y$ (the output layer of loss function!), then we have $\rho(\boldsymbol{x}, y; \theta) = \ell_y(\boldsymbol{x}) + \max_{\boldsymbol{x}' \in N(\boldsymbol{x}, \varepsilon)} \{|\ell_y(\boldsymbol{x}') - \ell_y(\boldsymbol{x})|\}$.*

We note that this loss function is a "surrogate" loss function for Madry et al.'s loss function because $\ell_y(\boldsymbol{x}) + \max_{\boldsymbol{x}' \in N(\boldsymbol{x}, \varepsilon)} \{|\ell_y(\boldsymbol{x}') - \ell_y(\boldsymbol{x})|\} \geq \ell_y(\boldsymbol{x}) + \max_{\boldsymbol{x}' \in N(\boldsymbol{x}, \varepsilon)} \{(\ell_y(\boldsymbol{x}') - \ell_y(\boldsymbol{x}))\} = \max_{\boldsymbol{x}' \in N(\boldsymbol{x}, \varepsilon)} \ell_y(\boldsymbol{x}')$. Therefore, even at such a trivial instantiation, robust attribution regularization provides interesting guarantees.

## 3.2 Distributional Robustness Model

A different but popular model for robust optimization is the distributional robustness model. In this case we consider a family of distributions $\mathcal{P}$, each of which is supposed to be a "slight variation" of a base distribution $P$. The goal of robust optimization is then that certain objective functions obtain stable values over this entire family. Here we apply the same underlying idea to the distributional

robustness model: One should get a small loss value over the base distribution $P$, and for any distribution $Q \in \mathcal{P}$, the IG-based *attributions* change only a little if we move from $P$ to $Q$. This is formalized as:

$$\underset{\theta}{\text{minimize}} \quad \underset{P}{\mathbb{E}}[\ell(P; \theta)] + \lambda \sup_{Q \in \mathcal{P}} \{W_{d_{\text{IG}}}(P, Q)\},$$

where the $W_{d_{\text{IG}}}(P, Q)$ is the Wasserstein distance between $P$ and $Q$ under a distance metric $d_{\text{IG}}$.[6] We use IG to highlight that this metric is related to integrated gradients.

We propose again $d_{\text{IG}}(\boldsymbol{z}, \boldsymbol{z}') = s(\text{IG}_{\boldsymbol{h}}^{\ell}(\boldsymbol{z}, \boldsymbol{z}'))$. We are particularly interested in the case where $\mathcal{P}$ is a Wasserstein ball around the base distribution $P$, using "perturbation" cost metric $c(\cdot)$. This gives regularization term $\lambda \, \mathbb{E}_{W_c(P,Q) \leq \rho} \sup\{W_{d_{\text{IG}}}(P, Q)\}$. An unsatisfying aspect of this objective, as one can observe now, is that $W_{d_{\text{IG}}}$ and $W_c$ can take two *different* couplings, while intuitively we want to use only one coupling to transport $P$ to $Q$. For example, this objective allows us to pick a coupling $M_1$ under which we achieve $W_{d_{\text{IG}}}$ (recall that Wasserstein distance is an infimum over couplings), and a different coupling $M_2$ under which we achieve $W_c$, but under $M_1 = (Z, Z')$, $\mathbb{E}_{z,z' \sim M_1}[c(z, z')] > \rho$, violating the constraint. This motivates the following modification:

$$\underset{\theta}{\text{minimize}} \quad \underset{P}{\mathbb{E}}[\ell(P; \theta)] + \lambda \sup_{Q; M \in \prod(P,Q)} \left\{ \underset{Z,Z'}{\mathbb{E}}[d_{\text{IG}}(Z, Z')] \text{ s.t. } \underset{Z,Z'}{\mathbb{E}}[c(Z, Z')] \leq \rho \right\}, \quad (5)$$

In this formulation, $\prod(P, Q)$ is the set of couplings of $P$ and $Q$, and $M = (Z, Z')$ is one coupling. $c(\cdot, \cdot)$ is a metric, such as $\|\cdot\|_2$, to measure the cost of an adversary perturbing $z$ to $z'$. $\rho$ is an upper bound on the expected perturbation cost, thus constraining $P$ and $Q$ to be "close" with each together. $d_{\text{IG}}$ is a metric to measure the change of attributions from $Z$ to $Z'$, where we want a large $d_{\text{IG}}$-change under a small $c$-change. The supremum is taken over $Q$ and $\prod(P, Q)$.

**Proposition 4 (Wasserstein prediction robustness).** *Let $s(\cdot)$ be the summation function and $\lambda = 1$, then for any curve $\gamma$ and any layer $\boldsymbol{h}$, (5) reduces to $\sup_{Q: W_c(P,Q) \leq \rho} \{\mathbb{E}_Q[\ell(Q; \theta)]\}$, which is the objective proposed by Sinha, Namhoong, and Duchi [SND18] for robust predictions.*

**Lagrange relaxation.** For any $\gamma \geq 0$, the Lagrange relaxation of (5) is

$$\underset{\theta}{\text{minimize}} \quad \left\{ \underset{P}{\mathbb{E}}[\ell(P; \theta)] + \lambda \sup_{Q; M \in \prod(P,Q)} \left\{ \underset{M=(Z,Z')}{\mathbb{E}} \left[d_{\text{IG}}(Z, Z') - \gamma c(Z, Z')\right] \right\} \right\} \quad (6)$$

where the supremum is taken over $Q$ (unconstrained) and all couplings of $P$ and $Q$, and we want to find a coupling under which IG attributions change a lot, while the perturbation cost from $P$ to $Q$ with respect to $c$ is small. Recall that $g: \mathbb{R}^d \times \mathbb{R}^d \to \mathbb{R}$ is a *normal integrand* if for each $\alpha$, the mapping $z \to \{z' | g(z, z') \leq \alpha\}$ is closed-valued and measurable [RW09].

Our next two theorems generalize the duality theory in [SND18] to a much larger, but natural, class of objectives.

**Theorem 1.** *Suppose $c(z, z) = 0$ and $d_{\text{IG}}(z, z) = 0$ for any $z$, and suppose $\gamma c(z, z') - d_{\text{IG}}(z, z')$ is a normal integrand. Then, $\sup_{Q; M \in \prod(P,Q)} \{\mathbb{E}_{M=(Z,Z')}[d_{\text{IG}}^\gamma(Z, Z')]\} = \mathbb{E}_{z \sim P}[\sup_{z'} \{d_{\text{IG}}^\gamma(z, z')\}]$. Consequently, we have (6) to be equal to the following:*

$$\underset{\theta}{\text{minimize}} \quad \underset{z \sim P}{\mathbb{E}} \left[ \ell(z; \theta) + \lambda \sup_{z'} \{d_{\text{IG}}(z, z') - \gamma c(z, z')\} \right] \quad (7)$$

The assumption $d_{\text{IG}}(z, z) = 0$ is true for what we propose, and $c(z, z) = 0$ is true for any typical cost such as $\ell_p$ distances. The normal integrand assumption is also very weak, e.g., it is satisfied when $d_{\text{IG}}$ is continuous and $c$ is closed convex.

Note that (7) and (4) are very similar, and so we use (4) for the rest the paper. Finally, given Theorem 1, we are also able to connect (5) and (7) with the following duality result:

**Theorem 2.** *Suppose $c(z, z) = 0$ and $d_{\text{IG}}(z, z) = 0$ for any $z$, and suppose $\gamma c(z, z') - d_{\text{IG}}(z, z')$ is a normal integrand. For any $\rho > 0$, there exists $\gamma \geq 0$ such that the optimal solutions of (7) are optimal for (5).*

### 3.3 One Layer Neural Networks

We now consider the special case of one-layer neural networks, where the loss function takes the form of $\ell(\boldsymbol{x}, y; \boldsymbol{w}) = g(-y\langle \boldsymbol{w}, \boldsymbol{x} \rangle)$, $\boldsymbol{w}$ is the model parameters, $\boldsymbol{x}$ is a feature vector, $y$ is a label, and $g$ is nonnegative. We take $s(\cdot)$ to be $\| \cdot \|_1$, which corresponds to a strong instantiation that does not allow attributions to cancel each other. Interestingly, we prove that for natural choices of $g$, this is however exactly Madry et al.'s objective [MMS+17a], which corresponds to $s(\cdot) = \mathsf{sum}(\cdot)$. That is, the strong ($s(\cdot) = \| \cdot \|_1$) and weak instantiations ($s(\cdot) = \mathsf{sum}(\cdot)$) coincide for one-layer neural networks. This thus says that for generalized linear classifiers, "robust interpretation" coincides with "robust predictions," and further with classic soft-margin training.

**Theorem 3.** *Suppose that $g$ is differentiable, non-decreasing, and convex. Then for $\lambda = 1$, $s(\cdot) = \| \cdot \|_1$, and $\ell_\infty$ neighborhood, (4) reduces to Madry et al.'s objective:*

$$\sum_{i=1}^{m} \max_{\| \boldsymbol{x}'_i - \boldsymbol{x}_i \|_\infty \leq \varepsilon} g(-y_i \langle \boldsymbol{w}, \boldsymbol{x}'_i \rangle) \ \textit{(Madry et al.'s objective)}$$

$$= \sum_{i=1}^{m} g(-y_i \langle \boldsymbol{w}, \boldsymbol{x}_i \rangle + \varepsilon \| \boldsymbol{w} \|_1) \ \textit{(soft-margin).}$$

Natural losses, such as Negative Log-Likelihood and softplus hinge loss, satisfy the conditions of this theorem.

## 4 Instantiations and Optimizations

In this section we discuss instantiations of (4) and how to optimize them. We start by presenting two objectives instantiated from our method: (1) IG-NORM, and (2) IG-SUM-NORM. Then we discuss how to use gradient descent to optimize these objectives.

**IG-NORM**. As our first instantiation, we pick $s(\cdot) = \| \cdot \|_1$, $\boldsymbol{h}$ to be the input layer, and $r$ to be the straightline connecting $\boldsymbol{x}$ and $\boldsymbol{x}'$. This gives:

$$\underset{\theta}{\text{minimize}} \quad \underset{(\boldsymbol{x}, y) \sim P}{\mathbb{E}} \left[ \ell(\boldsymbol{x}, y; \theta) + \lambda \max_{\boldsymbol{x}' \in N(\boldsymbol{x}, \varepsilon)} \| \mathrm{IG}^{\ell_y}(\boldsymbol{x}, \boldsymbol{x}') \|_1 \right]$$

**IG-SUM-NORM**. In the second instantiation we combine the sum size function and norm size function, and define $s(\cdot) = \mathsf{sum}(\cdot) + \beta \| \cdot \|_1$. Where $\beta \geq 0$ is a regularization parameter. Now with the same $\boldsymbol{h}$ and $r$ as above, and put $\lambda = 1$, then our method simplifies to:

$$\underset{\theta}{\text{minimize}} \quad \underset{(\boldsymbol{x}, y) \sim P}{\mathbb{E}} \left[ \max_{\boldsymbol{x}' \in N(\boldsymbol{x}, \varepsilon)} \left\{ \ell(\boldsymbol{x}', y; \theta) + \beta \| \mathrm{IG}^{\ell_y}(\boldsymbol{x}, \boldsymbol{x}') \|_1 \right\} \right]$$

which can be viewed as appending an extra robust IG term to $\ell(\boldsymbol{x}')$.

**Gradient descent optimization**. We propose the following gradient descent framework to optimize the objectives. The framework is parameterized by an adversary $\mathcal{A}$ which is supposed to solve the inner max by finding a point $\boldsymbol{x}^\star$ which changes attribution significantly. Specifically, given a point $(\boldsymbol{x}, y)$ at time step $t$ during SGD training, we have the following two steps (this can be easily generalized to mini-batches):

*Attack step*. We run $\mathcal{A}$ on $(\boldsymbol{x}, y)$ to find $\boldsymbol{x}^\star$ that produces a large inner max term (that is $\| \mathrm{IG}^{\ell_y}(\boldsymbol{x}, \boldsymbol{x}^\star) \|_1$ for IG-NORM, and $\ell(\boldsymbol{x}^\star) + \beta \| \mathrm{IG}^{\ell_y}(\boldsymbol{x}, \boldsymbol{x}^\star) \|_1$ for IG-SUM-NORM.

*Gradient step*. Fixing $\boldsymbol{x}^\star$, we can then compute the gradient of the corresponding objective with respect to $\theta$, and then update the model.

**Important objective parameters.** In both attack and gradient steps, we need to differentiate IG (in attack step, $\theta$ is fixed and we differentiate w.r.t. $\boldsymbol{x}$, while in gradient step, this is reversed), and this induces a set of parameters of the objectives to tune for optimization, which is summarized in Table 1. Differentiating summation approximation of IG amounts to compute second partial derivatives. We rely on the auto-differentiation capability of TensorFlow [ABC+16] to compute second derivatives.

| Adversary $\mathcal{A}$ | Adversary to find $\boldsymbol{x}^\star$. Note that our goal is simply to maximize the inner term in a neighborhood, thus in this paper we choose Projected Gradient Descent for this purpose. |
|---|---|
| $m$ in the attack step | To differentiate IG in the attack step, we use summation approximation of IG, and this is the number of segments for apprioximation. |
| $m$ in the gradient step | Same as above, but in the gradient step. We have this $m$ separately due to efficiency consideration. |
| $\lambda$ | Regularization parameter for IG-NORM. |
| $\beta$ | Regularization parameter for IG-SUM-NORM. |

Table 1: Optimization parameters.

# 5    Experiments

We now perform experiments using our method. We ask the following questions: **(1)** Comparing models trained by our method and naturally trained models at *test time*, do we maintain the accuracy on unperturbed test inputs? **(2)** At test time, if we use attribution attacks mentioned in [GAZ17] to perturb attributions while keeping predictions intact, how does the attribution robustness of our models compare with that of the naturally trained models? **(3)** Finally, how do we compare attribution robustness of our models with *weak instantiations* for robust predictions?

To answer these questions, We perform experiments on four classic datasets: MNIST [LCB98], Fashion-MNIST [XRV17], GTSRB [SSSI12], and Flower [NZ06]. In summary, our findings are the following: **(1)** Our method results in very small drop in test accuracy compared with naturally trained models. **(2)** On the other hand, our method gives signficantly better attribution robustness, as measured by correlation analyses. **(3)** Finally, our models yield *comparable* prediction robustness (sometimes even better), while *consistently* improving attribution robustness. In the rest of the section we give more details.

**Evaluation setup**. In this work we use IG to compute attributions (i.e. feature importance map), which, as demonstrated by [GAZ17], is more robust compared to other related methods (note that, IG also enjoys other theoretical properties). To attack attribution while retaining model predictions, we use Iterative Feature Importance Attacks (IFIA) proposed by [GAZ17]. Due to lack of space, we defer details of parameters and other settings to the appendix. We use two metrics to measure attribution robustness (i.e. how similar the attributions are between original and perturbed images):

***Kendall's tau rank order correlation***. Attribution methods rank all of the features in order of their importance, we thus use the rank correlation [Ken38] to compare similarity between interpretations.

***Top-k intersection***. We compute the size of intersection of the $k$ most important features before and after perturbation.

Compared with [GAZ17], we use Kendall's tau correlation, instead of Spearman's rank correlation. The reason is that we found that on the GTSRB and Flower datasets, Spearman's correlation is not consistent with visual inspection, and often produces too high correlations. In comparison, Kendall's tau correlation consistently produces lower correlations and aligns better with visual inspection. Finally, when computing attribution robustness, we only consider the test samples that are correctly classified by the model.

**Comparing with natural models**.    Figures (a), (b), (c), and (d) in Figure 2 show that, compared with naturally trained models, robust attribution training gives significant improvements in attribution robustness (measured by either median or confidence intervals). The exact numbers are recorded in Table 2: Compared with naturally trained models (rows where "Approach" is NATURAL), robust attribution training has significantly better adversarial accuracy and attribution robustness, while having a small drop in natural accuracy (denoted by `Nat Acc.`).

**Ineffective optimization**. We observe that even when our training stops, the attribution regularization term remains much more significant compared to the natural loss term. For example for IG-NORM, when training stops on MNIST, $\ell(\boldsymbol{x})$ typically stays at  1, but $\| \operatorname{IG}(\boldsymbol{x}, \boldsymbol{x}') \|_1$ stays at $10 \sim 20$. This indicates that optimization has not been very effective in minimizing the regularization term. There are two possible reasons to this: (1) Because we use summation approximation of IG, it forces us to compute second derivatives, which may not be numerically stable for deep net-

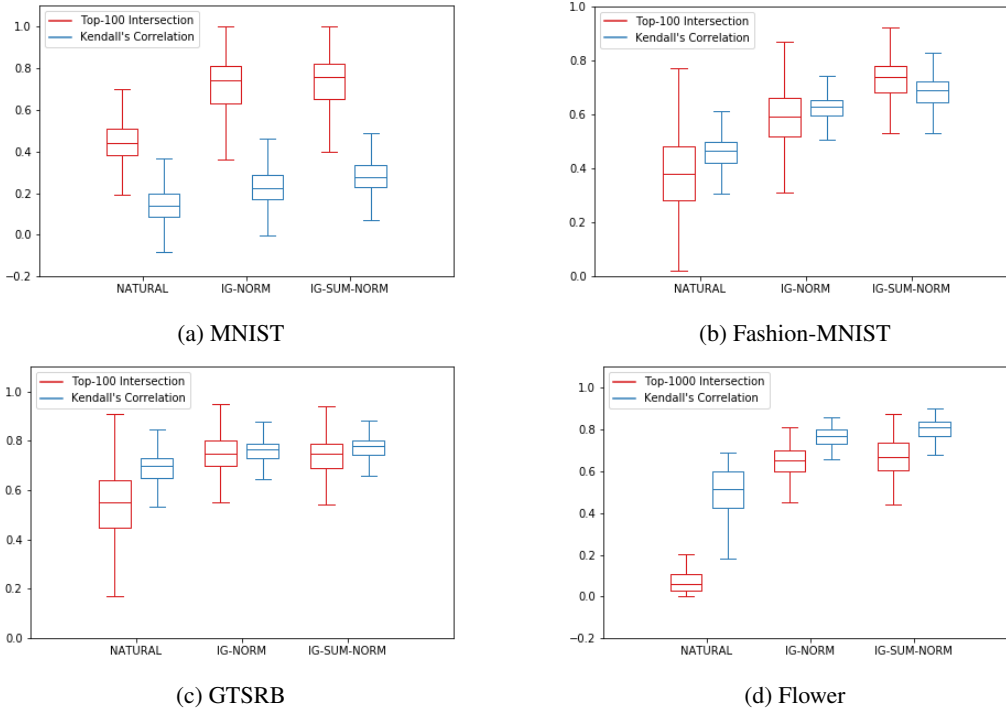

(a) MNIST

(b) Fashion-MNIST

(c) GTSRB

(d) Flower

Figure 2: Experiment results on MNIST, Fashion-MNIST, GTSRB and Flower.

works. (2) The network architecture may be inherently unsuitable for robust attributions, rendering the optimization hard to converge.

**Comparing with robust prediction models**. Finally we compare with Madry et al.'s models, which are trained for robust prediction. We use `Adv Acc.` to denote adversarial accuracy (prediction accuracy on perturbed inputs). Again, `TopK Inter.` denotes the average topK intersection ($K = 100$ for MNIST, Fashion-MNIST and GTSRB datasets, $K = 1000$ for Flower), and `Rank Corr.` denotes the average Kendall's rank order correlation. Table 2 gives the details of the results. As we can see, our models give comparable adversarial accuracy, and are sometimes even better (on the Flower dataset). On the other hand, we are consistently better in terms of attribution robustness.

| Dataset | Approach | Nat Acc. | Adv Acc. | TopK Inter. | Rank Corr. |
|---|---|---|---|---|---|
| MNIST | NATURAL | 99.17% | 0.00% | 46.61% | 0.1758 |
| | Madry et al. | 98.40% | 92.47% | 62.56% | 0.2422 |
| | IG-NORM | 98.74% | 81.43% | 71.36% | 0.2841 |
| | IG-SUM-NORM | 98.34% | 88.17% | **72.45%** | **0.3111** |
| Fashion-MNIST | NATURAL | 90.86% | 0.01% | 39.01% | 0.4610 |
| | Madry et al. | 85.73% | 73.01% | 46.12% | 0.6251 |
| | IG-NORM | 85.13% | 65.95% | 59.22% | 0.6171 |
| | IG-SUM-NORM | 85.44% | 70.26% | **72.08%** | **0.6747** |
| GTSRB | NATURAL | 98.57% | 21.05% | 54.16% | 0.6790 |
| | Madry et al. | 97.59% | 83.24% | 68.85% | 0.7520 |
| | IG-NORM | 97.02% | 75.24% | **74.81%** | 0.7555 |
| | IG-SUM-NORM | 95.68% | 77.12% | 74.04% | **0.7684** |
| Flower | NATURAL | 86.76% | 0.00% | 8.12% | 0.4978 |
| | Madry et al. | 83.82% | 41.91% | 55.87% | 0.7784 |
| | IG-NORM | 85.29% | 24.26% | 64.68% | 0.7591 |
| | IG-SUM-NORM | 82.35% | 47.06% | **66.33%** | **0.7974** |

Table 2: Experiment results including prediction accuracy, prediction robustness and attribution robustness.

# 6 Discussion and Conclusion

This paper builds a theory to robustify model interpretations through the lens of axiomatic attributions of neural networks. We show that our theory gives principled generalizations of previous formulations for robust predictions, and we characterize our objectives for one-layer neural networks. We believe that our work opens many intriguing avenues for future research, and we discuss a few topics below.

**Why we want robust attributions?** Model attributions are *facts* about model behaviors. While robust attribution does not necessarily mean that the attribution is correct, a model with *brittle attribution* can never be trusted. To this end, it seems interesting to examine attribution methods other than Integrated Gradients.

**Robust attribution leads to more human-aligned attribution.** Note that our proposed training scheme requires both prediction correctness and robust attributions, and therefore it encourages to learn *invariant* features from data that are also highly predictive. In our experiments, we found an intriguing phenomenon that *our regularized models produce attributions that are much more aligned with human perceptions* (for example, see Figure 1). Our results are aligned with the recent work [TSE+19, EIS+19].

**Robust attribution may help tackle spurious correlations.** In view of our discussion so far, we think it is plausible that robust attribution regularization can help remove spurious correlations because intuitively spurious correlations should not be able to be reliably attributed to. Future research on this potential connection seems warranted.

**Difficulty of optimization.** While our experimental results are encouraging, we observe that when training stops, the attribution regularization term remains significant (typically around tens to hundreds), which indicates ineffective optimization for the objectives. To this end, a main problem is network depth, where as depth increases, we get very unstable trajectories of gradient descent, which seems to be related to the use of *second order information* during robust attribution optimization (due to summation approximation, we have first order terms in the training objectives). Therefore, it is natural to further study better optimization tchniques or better architectures for robust attribution training.

# 7 Acknowledgments

This work is partially supported by Air Force Grant FA9550-18-1-0166, the National Science Foundation (NSF) Grants CCF-FMitF-1836978, SaTC-Frontiers-1804648 and CCF-1652140 and ARO grant number W911NF-17-1-0405.

## Footnotes

[4] Proofs are deferred to B.2.

[5] We stress that this regularization term depends on model parameters $\theta$ through loss function $\ell_y$.

[6] For supervised learning problem where $P$ is of the form $Z = (X, Y)$, we use the same treatment as in [SND18] so that cost function is defined as $c(z, z') = c_x(x, x') + \infty \cdot \mathbf{1}\{y \neq y'\}$. All our theory carries over to such $c$ which has range $\mathbb{R}_+ \cup \{\infty\}$.

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
