[Supplementary Material]

## A    Code

Code for this paper is publicly available at the following repository:

## B    Proofs

### B.1    Additional definitions

Let $P, Q$ be two distributions, a coupling $M = (Z, Z')$ is a joint distribution, where, if we marginalize $M$ to the first component, $Z$, it is identically distributed as $P$, and if we marginalize $M$ to the second component, $Z'$, it is identically distributed as $Q$. Let $\prod(P, Q)$ be the set of all couplings of $P$ and $Q$, and let $c(\cdot, \cdot)$ be a "cost" function that maps $(z, z')$ to a real value. Wasserstein distance between $P$ and $Q$ w.r.t. $c$ is defined as

$$W_c(P, Q) = \inf_{M \in \prod(P,Q)} \left\{ \mathbb{E}_{(z,z') \sim M} [c(z, z')] \right\}.$$

Intuitively, this is to find the "best transportation plan" (a coupling $M$) to minimize the expected transportation cost (transporting $z$ to $z'$ where the cost is $c(z, z')$).

### B.2    Integrated Gradients for an Intermediate Layer

In this section we show how to compute Integrated Gradients for an intermediate layer of a neural network. Let $h : \mathbb{R}^d \mapsto \mathbb{R}^k$ be a function that computes a hidden layer of a neural network, where we map a $d$-dimensional input vector to a $k$-dimensional output vector. Given two points $\boldsymbol{x}$ and $\boldsymbol{x}'$ for computing attribution, again we consider a parameterization (which is a mapping $r : \mathbb{R} \mapsto \mathbb{R}^d$) such that $r(0) = \boldsymbol{x}$, and $r(1) = \boldsymbol{x}'$.

The key insight is to leverage the fact that Integrated Gradients is a *curve integration*. Therefore, given some hidden layer, one can then naturally view the previous layers as inducing a *curve* $h \circ r$ which moves from $h(\boldsymbol{x})$ to $h(\boldsymbol{x}')$, as we move from $\boldsymbol{x}$ to $\boldsymbol{x}'$ along curve $r$. Viewed this way, we can thus naturally compute IG for $\boldsymbol{h}$ in a way that leverages all layers of the network. Specifically, consider another curve $\gamma(t) : \mathbb{R} \mapsto \mathbb{R}^k$, defined as $\gamma(t) = h(r(t))$, to compute a curve integral. By definition we have $f(\boldsymbol{x}) = g(h(\boldsymbol{x}))$

$$f(\boldsymbol{x}') - f(\boldsymbol{x}) = g(h(\boldsymbol{x}')) - g(h(\boldsymbol{x}))$$
$$= g(\gamma(1)) - g(\gamma(0))$$
$$= \int_0^1 \sum_{i=1}^k \frac{\partial f(\gamma(t))}{\partial h_i} \gamma_i'(t) dt$$
$$= \sum_{i=1}^k \int_0^1 \frac{\partial f(\gamma(t))}{\partial h_i} \gamma_i'(t) dt$$

Therefore we can define the attribution of $h_i$ naturally as

$$\text{IG}_{h_i}^f(\boldsymbol{x}, \boldsymbol{x}') = \int_0^1 \frac{\partial f(\gamma(t))}{\partial h_i} \gamma_i'(t) dt$$

Let's unpack this a little more:

$$\int_0^1 \frac{\partial f(\gamma(t))}{\partial h_i} \gamma_i'(t) dt = \int_0^1 \frac{\partial f(h(r(t)))}{\partial h_i} \sum_{j=1}^d \frac{\partial h_i(r(t))}{\partial \boldsymbol{x}_j} r_j'(t) dt$$
$$= \int_0^1 \frac{\partial f(h(r(t)))}{\partial h_i} \sum_{j=1}^d \frac{\partial h_i(r(t))}{\partial \boldsymbol{x}_j} r_j'(t) dt$$
$$= \sum_{j=1}^d \left\{ \int_0^1 \frac{\partial f(h(r(t)))}{\partial h_i} \frac{\partial h_i(r(t))}{\partial \boldsymbol{x}_j} r_j'(t) dt \right\}$$

This thus gives the lemma

**Lemma 2.** *Under curve* $r : \mathbb{R} \mapsto \mathbb{R}^d$ *where* $r(0) = \boldsymbol{x}$ *and* $r(1) = \boldsymbol{x}'$, *the attribution for* $h_i$ *for a differentiable function* $f$ *is*

$$\text{IG}_{h_i}^f(\boldsymbol{x}, \boldsymbol{x}', r) = \sum_{j=1}^d \left\{ \int_0^1 \frac{\partial f(h(r(t)))}{\partial h_i} \frac{\partial h_i(r(t))}{\partial \boldsymbol{x}_j} r_j'(t) dt \right\} \tag{8}$$

Note that (6) nicely recovers attributions for input layer, in which case $h$ is the identity function.

**Summation approximation.** Similarly, we can approximate the above Riemann integral using a summation. Suppose we slice $[0, 1]$ into $m$ equal segments, then (2) can be approximated as:

$$\text{IG}_{h_i}^f(\boldsymbol{x}, \boldsymbol{x}') = \frac{1}{m} \sum_{j=1}^d \left\{ \sum_{k=0}^{m-1} \frac{\partial f(h(r(k/m)))}{\partial h_i} \frac{\partial h_i(r(k/m))}{\partial \boldsymbol{x}_j} r_j'(k/m) \right\} \tag{9}$$

### B.3   Proof of Proposition 1

If we put $\lambda = 1$ and let $s(\cdot)$ be the sum function (sum all components of a vector), then for any curve $r$ and any intermediate layer $\boldsymbol{h}$, (4) becomes:

$$\begin{aligned}
\rho(\boldsymbol{x}, y; \theta) &= \ell(\boldsymbol{x}, y; \theta) + \max_{\boldsymbol{x}' \in N(\boldsymbol{x}, \varepsilon)} \{\text{sum}(\text{IG}^{\ell_y}(\boldsymbol{x}, \boldsymbol{x}'; r))\} \\
&= \ell(\boldsymbol{x}, y; \theta) + \max_{\boldsymbol{x}' \in N(\boldsymbol{x}, \varepsilon)} \{\ell(\boldsymbol{x}', y; \theta) - \ell(\boldsymbol{x}, y; \theta)\} \\
&= \max_{\boldsymbol{x}' \in N(\boldsymbol{x}, \varepsilon)} \ell(\boldsymbol{x}', y; \theta)
\end{aligned}$$

where the second equality is due to the Axiom of Completeness of IG.

### B.4   Proof of Proposition 2

Input gradient regularization is an old idea proposed by Drucker and LeCun [DL92], and is recently used by Ross and Doshi-Velez [RD18] in adversarial training setting. Basically, for $q \geq 1$, they propose $\rho(\boldsymbol{x}, y; \theta) = \ell(\boldsymbol{x}, y; \theta) + \lambda \|\nabla_{\boldsymbol{x}} \ell(\boldsymbol{x}, y; \theta)\|_q^q$, where they want small gradient at $\boldsymbol{x}$. To recover this objective from robust attribution regularization, let us pick $s(\cdot)$ as the $\|\cdot\|_1^q$ function (1-norm to the $q$-th power), and consider the simplest curve $r(t) = \boldsymbol{x} + t(\boldsymbol{x}' - \boldsymbol{x})$. With the naïve summation approximation of the integral $\text{IG}_i^{\ell_y}$ we have $\text{IG}_i^{\ell_y}(\boldsymbol{x}, \boldsymbol{x}'; r) \approx \frac{(\boldsymbol{x}_i' - \boldsymbol{x}_i)}{m} \sum_{k=1}^m \frac{\partial \ell(\boldsymbol{x} + \frac{k-1}{m}(\boldsymbol{x}' - \boldsymbol{x}), y; \theta)}{\partial \boldsymbol{x}_i}$, where larger $m$ is, more accurate we approximate the integral. Now, if we put $m = 1$, which is the coarsest approximation, this becomes $(\boldsymbol{x}_i' - \boldsymbol{x}_i) \frac{\partial \ell(\boldsymbol{x}, y; \theta)}{\partial \boldsymbol{x}_i}$, and we have $\text{IG}^{\ell_y}(\boldsymbol{x}, \boldsymbol{x}'; \theta) = (\boldsymbol{x}' - \boldsymbol{x}) \odot \nabla_{\boldsymbol{x}} \ell(\boldsymbol{x}, y; \theta)$. Therefore (4) becomes:

$$\begin{aligned}
\rho(\boldsymbol{x}, y; \theta) &= \ell(\boldsymbol{x}, y; \theta) + \lambda \max_{\boldsymbol{x}' \in N(\boldsymbol{x}, \varepsilon)} \{\|\text{IG}^{\ell_y}(\boldsymbol{x}, \boldsymbol{x}'; \theta)\|_1^q\} \\
&\approx \ell(\boldsymbol{x}, y; \theta) + \lambda \max_{\boldsymbol{x}' \in N(\boldsymbol{x}, \varepsilon)} \{\|(\boldsymbol{x}' - \boldsymbol{x}) \odot \nabla_{\boldsymbol{x}} \ell(\boldsymbol{x}, y; \theta)\|_1^q\}
\end{aligned}$$

Put the neighborhood as $\|\boldsymbol{x}' - \boldsymbol{x}\|_p \leq \varepsilon$ where $p \in [1, \infty]$ and $\frac{1}{p} + \frac{1}{q} = 1$. By Hölder's inequality, $\|(\boldsymbol{x}' - \boldsymbol{x}) \odot \nabla_{\boldsymbol{x}} \ell(\boldsymbol{x}, y; \theta)\|_1^q \leq \|\boldsymbol{x}' - \boldsymbol{x}\|_p^q \|\nabla \ell(\boldsymbol{x}, y; \theta)\|_q^q \leq \varepsilon^q \|\nabla \ell(\boldsymbol{x}, y; \theta)\|_q^q$ which means that $\max_{\|\boldsymbol{x}' - \boldsymbol{x}\|_p \leq \varepsilon} \{\|(\boldsymbol{x}' - \boldsymbol{x}) \odot \nabla_{\boldsymbol{x}} \ell(\boldsymbol{x}, y; \theta)\|_1^q\} = \varepsilon^q \|\nabla \ell(\boldsymbol{x}, y; \theta)\|_q^q$. Thus by putting $\lambda = \lambda'/\varepsilon^q$, we recover gradient regularization with regularization parameter $\lambda'$.

### B.5   Proof of Proposition 3

Let us put $s(\cdot) = \|\cdot\|_1$, and $\boldsymbol{h} = \ell_y$ (the output layer of loss function!), then we have

$$\begin{aligned}
\rho(\boldsymbol{x}, y; \theta) &= \ell_y(\boldsymbol{x}) + \max_{\boldsymbol{x}' \in N(\boldsymbol{x}, \varepsilon)} \{\|\text{IG}_{\ell_y}^{\ell_y}(\boldsymbol{x}, \boldsymbol{x}'; r)\|_1\} \\
&= \ell_y(\boldsymbol{x}) + \max_{\boldsymbol{x}' \in N(\boldsymbol{x}, \varepsilon)} \{|\ell_y(\boldsymbol{x}') - \ell_y(\boldsymbol{x})|\}
\end{aligned}$$

where the second equality is because $\text{IG}_{\ell_y}^{\ell_y}(\boldsymbol{x}, \boldsymbol{x}'; r) = \ell_y(\boldsymbol{x}') - \ell_y(\boldsymbol{x})$.

## B.6 Proof of Proposition 4

Specifically, again, let $s(\cdot)$ be the summation function and $\lambda = 1$, then we have $\mathbb{E}_{Z,Z'}[d_{\mathrm{IG}}(Z,Z')] = \mathbb{E}_{Z,Z'}[\mathrm{sum}(\mathrm{IG}_{\boldsymbol{h}}^{\ell}(Z,Z'))] = \mathbb{E}_{Z,Z'}[\ell(Z';\theta) - \ell(Z;\theta)]$. Because $P$ and $Z$ are identically distributed, thus the objective reduces to

$$\sup_{Q;M\in\prod(P,Q)} \left\{ \mathbb{E}_{Z,Z'}[\ell(Z;\theta) + \ell(Z';\theta) - \ell(Z;\theta)] \right.$$

$$\left. \text{s.t. } \mathbb{E}_{Z,Z'}[c(Z,Z')] \leq \rho \right\}$$

$$= \sup_{Q;M\in\prod(P,Q)} \left\{ \mathbb{E}_{Z'}[\ell(Z';\theta)] \text{ s.t. } \mathbb{E}_{Z,Z'}[c(Z,Z')] \leq \rho \right\}$$

$$= \sup_{Q:W_c(P,Q)\leq\rho} \left\{ \mathbb{E}_{Q}[\ell(Q;\theta)] \right\},$$

which is exactly Wasserstein prediction robustness objective.

## B.7 Proof of Theorem 1

The proof largely follows that for Theorem 5 in [SND18], and we provide it here for completeness. Since we have a joint supremum over $Q$ and $M \in \prod(P,Q)$ we have that

$$\sup_{Q;M\in\prod(P,Q)} \left\{ \mathbb{E}_{M=(Z,Z')}[d_{\mathrm{IG}}^{\gamma}(Z,Z')] \right\} = \sup_{Q;M\in\prod(P,Q)} \int [d_{\mathrm{IG}}(z,z') - \gamma c(z,z')]dM(z,z')$$

$$\leq \int \sup_{z'}\{d_{\mathrm{IG}}(z,z') - \gamma c(z,z')\}dP(z)$$

$$= \mathbb{E}_{z\sim P}\left[\sup_{z'}\{d_{\mathrm{IG}}^{\gamma}(z,z')\}\right].$$

We would like to show equality in the above.

Let $\mathcal{Q}$ denote the space of regular conditional probabilities from $Z$ to $Z'$. Then

$$\sup_{Q;M\in\prod(P,Q)} \int [d_{\mathrm{IG}}(z,z') - \gamma c(z,z')]dM(z,z') \geq \sup_{Q\in\mathcal{Q}} \int [d_{\mathrm{IG}}(z,z') - \gamma c(z,z')]dQ(z'|z)dP(z).$$

Let $\mathcal{Z}'$ denote all measurable mappings $z \to z'(z)$ from $Z$ to $Z'$. Using the measurability result in Theorem 14.60 in [RW09], we have

$$\sup_{z'\in\mathcal{Z}'} \int [d_{\mathrm{IG}}(z,z'(z)) - \gamma c(z,z'(z))]dP(z) = \int \sup_{z'}[d_{\mathrm{IG}}(z,z') - \gamma c(z,z')]dP(z)$$

since $\gamma c - d_{\mathrm{IG}}$ is a normal integrand.

Let $z'(z)$ be any measurable function that is $\epsilon$-close to attaining the supremum above, and define the conditional distribution $Q(z'|z)$ to be supported on $z'(z)$. Then

$$\sup_{Q;M\in\prod(P,Q)} \int [d_{\mathrm{IG}}(z,z') - \gamma c(z,z')]dM(z,z') \geq \int [d_{\mathrm{IG}}(z,z') - \gamma c(z,z')]dQ(z'|z)dP(z)$$

$$= \int [d_{\mathrm{IG}}(z,z'(z)) - \gamma c(z,z'(z))]dP(z)$$

$$\geq \int \sup_{z'}[d_{\mathrm{IG}}(z,z') - \gamma c(z,z')]dP(z) - \epsilon$$

$$\geq \sup_{Q;M\in\prod(P,Q)} \int [d_{\mathrm{IG}}(z,z') - \gamma c(z,z')]dM(z,z') - \epsilon.$$

Since $\epsilon \geq 0$ is arbitrary, this completes the proof. $\qquad\square$

## B.8 Proof of Theorem 2: Connections Between the Distributional Robustness Objectives

Let $\theta^*$ denote an optimal solution of (5) and let $\theta'$ be any non-optimal solution. Let $\gamma(\theta^*)$ denote the corresponding $\gamma$ by Lemma 3, and $\gamma(\theta')$ denote that for $\theta'$.

Since $\gamma(\theta')$ achieves the infimum, we have

$$\mathbb{E}_{z \sim P}\left[\ell(z; \theta') + \lambda \sup_{z'}\{d_{\mathrm{IG}}(z, z') - \gamma(\theta^*)c(z, z')\}\right] \tag{10}$$

$$\geq \mathbb{E}_{z \sim P}\left[\ell(z; \theta') + \lambda \sup_{z'}\{d_{\mathrm{IG}}(z, z') - \gamma(\theta')c(z, z')\}\right] \tag{11}$$

$$> \mathbb{E}_{z \sim P}\left[\ell(z; \theta^*) + \lambda \sup_{z'}\{d_{\mathrm{IG}}(z, z') - \gamma(\theta^*)c(z, z')\}\right]. \tag{12}$$

So $\theta'$ is not optimal for (7). This then completes the proof. □

**Lemma 3.** *Suppose $c(z, z) = 0$ and $d_{\mathrm{IG}}(z, z) = 0$ for any z, and suppose $\gamma c(z, z') - d_{\mathrm{IG}}(z, z')$ is a normal integrand. For any $\rho > 0$, there exists $\gamma \geq 0$ such that*

$$\sup_{Q; M \in \prod(P, Q)}\left\{\mathbb{E}_{(Z, Z') \sim M}[d_{\mathrm{IG}}(Z, Z')] \; s.t. \; \mathbb{E}_{(Z, Z') \sim M}[c(Z, Z')] \leq \rho\right\} \tag{13}$$

$$= \inf_{\zeta \geq 0} \mathbb{E}_{z \sim P}\left[\sup_{z'}\{d_{\mathrm{IG}}(z, z') - \zeta c(z, z') + \zeta \rho\}\right]. \tag{14}$$

*Furthermore, there exists $\gamma \geq 0$ achieving the infimum.*

This lemma generalizes Theorem 5 in [SND18] to a larger, but natural, class of objectives.

*Proof.* For $Q$ and $M \in \Pi(P, Q)$, let

$$\Lambda_{\mathrm{IG}}(Q, M) := \mathbb{E}_{(Z, Z') \sim M}[d_{\mathrm{IG}}(Z, Z')] \tag{15}$$

$$\Lambda_c(Q, M) := \mathbb{E}_{(Z, Z') \sim M}[c(Z, Z')] \tag{16}$$

First, the pair $(Q, M)$ forms a convex set, and $\Lambda_{\mathrm{IG}}(Q, M)$ and $\Lambda_c(Q, M)$ are linear functionals over the convex set. Set $Q = P$ and set $M$ to the identity coupling (such that $(Z, Z') \sim M$ always has $Z = Z'$). Then $\Lambda_c(Q, M) = 0 < \rho$ and thus the Slater's condition holds. Applying standard infinite dimensional duality results (Theorem 8.6.1 in [Lue97]) leads to

$$\sup_{Q; M \in \prod(P, Q); \Lambda_c(Q, M) \leq \rho} \Lambda_{\mathrm{IG}}(Q, M) \tag{17}$$

$$= \sup_{Q; M \in \prod(P, Q)} \inf_{\zeta \geq 0}\{\Lambda_{\mathrm{IG}}(Q, M) - \zeta \Lambda_c(Q, M) + \zeta \rho\} \tag{18}$$

$$= \inf_{\zeta \geq 0} \sup_{Q; M \in \prod(P, Q)}\{\Lambda_{\mathrm{IG}}(Q, M) - \zeta \Lambda_c(Q, M) + \zeta \rho\}. \tag{19}$$

Furthermore, there exists $\gamma \geq 0$ achieving the infimum in the last line.

Now, it suffices to show that

$$\sup_{Q; M \in \prod(P, Q)}\{\Lambda_{\mathrm{IG}}(Q, M) - \gamma \Lambda_c(Q, M) + \gamma \rho\} \tag{20}$$

$$= \mathbb{E}_{z \sim P}\left[\sup_{z'}\{d_{\mathrm{IG}}(z, z') - \gamma c(z, z') + \gamma \rho\}\right]. \tag{21}$$

This is exactly what Theorem 1 shows. □

## B.9 Proof of Theorem 3

Let us fix any one point $\boldsymbol{x}$, and consider $g(-y_i\langle \boldsymbol{w}, \boldsymbol{x}\rangle) + \lambda \max_{\boldsymbol{x}' \in N(\boldsymbol{x}, \varepsilon)} \|\mathrm{IG}_{\boldsymbol{x}}^{\ell_y}(\boldsymbol{x}, \boldsymbol{x}'; \boldsymbol{w})\|_1$. Due to the special form of $g$, we know that:

$$\mathrm{IG}_i^{\ell_y}(\boldsymbol{x}, \boldsymbol{x}'; \boldsymbol{w}) = \frac{\boldsymbol{w}_i(\boldsymbol{x}' - \boldsymbol{x})_i}{\langle \boldsymbol{w}, \boldsymbol{x}' - \boldsymbol{x}\rangle} \cdot \big(g(-y\langle \boldsymbol{w}, \boldsymbol{x}'\rangle) - g(-y\langle \boldsymbol{w}, \boldsymbol{x}\rangle)\big)$$

Let $\Delta = \boldsymbol{x}' - \boldsymbol{x}$ (which satisfies that $\|\Delta\|_\infty \leq \varepsilon$), therefore its absolute value (note that we are taking 1-norm):

$$\frac{\left|g(-y\langle\boldsymbol{w},\boldsymbol{x}\rangle - y\langle\boldsymbol{w},\Delta\rangle) - g(-y\langle\boldsymbol{w},\boldsymbol{x}\rangle)\right|}{|\langle\boldsymbol{w},\Delta\rangle|} \cdot |\boldsymbol{w}_i\,\Delta_i|$$

Let $z = -y\langle\boldsymbol{w},\boldsymbol{x}\rangle$ and $\delta = -y\langle\boldsymbol{w},\Delta\rangle$, this is further simplified as $\frac{|g(z+\delta)-g(z)|}{|\delta|}|\delta_i|$. Because $g$ is non-decreasing, so $g' \geq 0$, and so this is indeed $\frac{g(z+\delta)-g(z)}{\delta}$, which is the slope of the secant from $(z, g(z))$ to $(z+\delta, g(z+\delta))$. Because $g$ is convex so the secant slopes are non-decreasing, so we can simply pick $\Delta_i = -y\,\mathrm{sgn}(\boldsymbol{w}_i)\varepsilon$, and so $\delta = \|\boldsymbol{w}\|_1\varepsilon$, and so that $\|\,\mathrm{IG}\,\|_1$ becomes

$$|g(z+\varepsilon\|\boldsymbol{w}\|_1) - g(z)| \cdot \frac{\sum_i|\boldsymbol{w}_i\,\Delta_i|}{|\delta|} = |g(z+\varepsilon\|\boldsymbol{w}\|_1) - g(z)| \cdot \frac{\sum_i|\boldsymbol{w}_i|\varepsilon}{\|\boldsymbol{w}\|_1\varepsilon}$$
$$= |g(z+\varepsilon\|\boldsymbol{w}\|_1) - g(z)|$$
$$= g(z+\varepsilon\|\boldsymbol{w}\|_1) - g(z)$$

where the last equality follows because $g$ is nondecreasing. Therefore the objective simplifies to $\sum_{i=1}^m g(-y_i\langle\boldsymbol{w},\boldsymbol{x}_i\rangle + \varepsilon\|\boldsymbol{w}\|_1)$, which is exactly Madry et al.'s objective under $\ell_\infty$ perturbations.  $\square$

Let us consider two examples:

*Logistic Regression.* Let $g(z) = \ln(1 + \exp(z))$. Then $g(-y\langle\boldsymbol{w},\boldsymbol{x}\rangle)$ recovers the Negative Log-Likelihood loss for logistic regression. Clearly $g$ is nondecreasing and $g'$ is also nondecreasing. As a result, adversarial training for logistic regression is exactly "robustifying" attributions/explanations.

*Softplus hinge loss.* Alternatively, we can let $g(z) = \ln(1 + \exp(1 + z))$, and therefore $g(-y\langle\boldsymbol{w},\boldsymbol{x}\rangle) = \ln(1 + \exp(1 - y\langle\boldsymbol{w},\boldsymbol{x}\rangle))$ is the softplus version of the hinge loss function. Clearly this $g$ also satisfy our requirements, and therefore adversarial training for softplus hinge loss function is also exactly about "robustifying" attributions/explanations.

## C  More Details of Experiments

### C.1  Experiment Settings

We perform experiments on four datasets: MNIST, Fashion-MNIST, GTSRB and Flower. Robust attribution regularization training requires extensive computing power. We conducted experiments in parallel over multiple NVIDIA Tesla V100 and NVDIA GeForce RTX 2080Ti GPUs both on premises and on cloud. Detailed experiment settings for each dataset are described below.

### C.1.1  MNIST

**Data**. The MNIST dataset [LCB98] is a large dataset of handwritten digits. Each digit has 5,500 training images and 1,000 test images. Each image is a $28 \times 28$ grayscale. We normalize the range of pixel values to $[0, 1]$.

**Model**. We use a network consisting of two convolutional layers with 32 and 64 filters respectively, each followed by $2 \times 2$ max-pooling, and a fully connected layer of size 1024. Note that we use the same MNIST model as [MMS$^+$17b].

**Training hyper-parameters**. The hyper-parameters to train different models are listed below:

*NATURAL.* We set learning rate as $10^{-4}$, batch size as 50, training steps as 25,000, and use Adam Optimizer.

*Madry et al..* We set learning rate as $10^{-4}$, batch size as 50, training steps as 100,000, and use Adam Optimizer. We use PGD attack as adversary with random start, number of steps of 40, step size of 0.01, and adversarial budget $\epsilon$ of 0.3.

*IG-NORM.* We set $\lambda = 1$, $m = 50$ for gradient step, learning rate as $10^{-4}$, batch size as 50, training steps as 100,000, and use Adam Optimizer. We use PGD attack as adversary with random start, number of steps of 40, step size of 0.01, $m = 10$ for attack step, and adversarial budget $\epsilon = 0.3$.

*IG-SUM-NORM.* We set $\beta$ as 0.1, $m$ in the gradient step as 50, learning rate as $10^{-4}$, batch size as 50, training steps as 100,000, and use Adam Optimizer. We use PGD attack as adversary with

random start, number of steps of 40, step size of 0.01, $m = 10$ in the attack step, and adversarial budget $\epsilon = 0.3$.

**Evaluation Attacks**. For attacking inputs to change model predictions, we use PGD attack with random start, number of steps of 100, adversarial budget $\epsilon$ of 0.3 and step size of 0.01. For attacking inputs to change interpretations, we use Iterative Feature Importance Attacks (IFIA) proposed by [GAZ17]. We use their top-k attack with $k = 200$, adversarial budget $\epsilon = 0.3$, step size $\alpha = 0.01$ and number of iterations $P = 100$. We set the feature importance function as Integrated Gradients(IG) and dissimilarity function $D$ as Kendall's rank order correlation. We find that IFIA is not stable if we use GPU parallel computing (non-deterministic is a behavior of GPU), so we run IFIA three times on each test example and use the best result with the lowest Kendall's rank order correlation.

### C.1.2 Fashion-MNIST

**Data**. The Fashion-MNIST dataset [XRV17] contains images depicting wearables such as shirts and boots instead of digits, which is more complex than MNIST dataset. The image format, the number of classes, as well as the number of examples are all identical to MNIST.

**Model**. We use a network consisting of two convolutional layers with 32 and 64 filters respectively, each followed by $2 \times 2$ max-pooling, and a fully connected layer of size 1024.

**Training hyper-parameters**. The hyper-parameters to train different models are listed below:

*NATURAL*. We set learning rate as $10^{-4}$, batch size as 50, training steps as 25,000, and use Adam Optimizer.

*Madry et al.*. We set learning rate as $10^{-4}$, batch size as 50, training steps as 100,000, and use Adam Optimizer. We use PGD attack as adversary with random start, number of steps of 20, step size of 0.01, and adversarial budget $\epsilon$ of 0.1.

*IG-NORM*. We set $\lambda = 1$, $m = 50$ for gradient step, learning rate as $10^{-4}$, batch size as 50, training steps as 100,000, and use Adam Optimizer. We use PGD attack as adversary with random start, number of steps of 20, step size of 0.01, $m = 10$ for attack step, and adversarial budget $\epsilon = 0.1$.

*IG-SUM-NORM*. We set $\beta$ as 0.1, $m$ in the gradient step as 50, learning rate as $10^{-4}$, batch size as 50, training steps as 100,000, and use Adam Optimizer. We use PGD attack as adversary with random start, number of steps of 20, step size of 0.01, $m = 10$ in the attack step, and adversarial budget $\epsilon = 0.1$.

**Evaluation Attacks**. For attacking inputs to change model predictions, we use PGD attack with random start, number of steps of 100, adversarial budget $\epsilon$ of 0.1 and step size of 0.01. For attacking inputs to change interpretations, we use Iterative Feature Importance Attacks (IFIA) proposed by [GAZ17]. We use their top-k attack with $k = 100$, adversarial budget $\epsilon = 0.1$, step size $\alpha = 0.01$ and number of iterations $P = 100$. We set the feature importance function as Integrated Gradients(IG) and dissimilarity function $D$ as Kendall's rank order correlation. We find that IFIA is not stable if we use GPU parallel computing (non-deterministic is a behavior of GPU), so we run IFIA three times on each test example and use the best result with the lowest Kendall's rank order correlation.

### C.1.3 GTSRB

**Data**. The German Traffic Sign Recognition Benchmark (GTSRB) [SSSI12] is a dataset of color images depicting 43 different traffic signs. The images are not of a fixed dimensions and have rich background and varying light conditions as would be expected of photographed images of traffic signs. There are about 34,799 training images, 4,410 validation images and 12,630 test images. We resize each image to $32 \times 32$. The pixel values are in range of $[0, 255]$. The dataset has a large imbalance in the number of sample occurrences across classes. We use data augmentation techniques to enlarge the training data and make the number of samples in each class balanced. We construct a class preserving data augmentation pipeline consisting of rotation, translation, and projection transforms and apply this pipeline to images in the training set until each class contained 10,000 training examples. We use this new augmented training data set containing 430,000 samples in total to train models. We also preprocess images via image brightness normalization.

**Model** . We use the Resnet model [HZRS16]. We perform per image standardization before feeding images to the neural network. The network has 5 residual units with (16, 16, 32, 64) filters each. The model is adapted from CIFAR-10 model of [MMS$^+$17b]. Refer to our codes for details.

**Training hyper-parameters**. The hyper-parameters to train different models are listed below:

*NATURAL*. We use Momentum Optimizer with weight decay. We set momentum rate as 0.9, weight decay rate as 0.0002, batch size as 64, and training steps as 70,000. We use learning rate schedule: the first 500 steps, we use learning rate of $10^{-3}$; after 500 steps and before 60,000 steps, we use learning rate of $10^{-2}$; after 60,000 steps, we use learning rate of $10^{-3}$.

*Madry et al.*. We use Momentum Optimizer with weight decay. We set momentum rate as 0.9, weight decay rate as 0.0002, batch size as 64, and training steps as 70,000. We use learning rate schedule: the first 500 steps, we use learning rate of $10^{-3}$; after 500 steps and before 60,000 steps, we use learning rate of $10^{-2}$; after 60,000 steps, we use learning rate of $10^{-3}$. We use PGD attack as adversary with random start, number of steps of 7, step size of 2, and adversarial budget $\epsilon$ of 8.

*IG-NORM*. We set $\lambda$ as 1, $m$ in the gradient step as 50. We use Momentum Optimizer with weight decay. We set momentum rate as 0.9, weight decay rate as 0.0002, batch size as 64, and training steps as 70,000. We use learning rate schedule: the first 500 steps, we use learning rate of $10^{-6}$; after 500 steps and before 60,000 steps, we use learning rate of $10^{-4}$; after 60,000 steps, we use learning rate of $10^{-5}$. We use PGD attack as adversary with random start, number of steps of 7, step size of 2, $m$ in the attack step of 5, and adversarial budget $\epsilon$ of 8.

*IG-SUM-NORM*. We set $\beta$ as 1, $m$ in the gradient step as 50. We use Momentum Optimizer with weight decay. We set momentum rate as 0.9, weight decay rate as 0.0002, batch size as 64, and training steps as 70,000. We use learning rate schedule: the first 500 steps, we use learning rate of $10^{-5}$; after 500 steps and before 60,000 steps, we use learning rate of $10^{-4}$; after 60,000 steps, we use learning rate of $10^{-5}$. We use PGD attack as adversary with random start, number of steps of 7, step size of 2, $m$ in the attack step of 5, and adversarial budget $\epsilon$ of 8.

**Evaluation Attacks**. For attacking inputs to change model predictions, we use PGD attack with number of steps of $40$, adversarial budget $\epsilon$ of 8 and step size of 2. For attacking inputs to change interpretations, we use Iterative Feature Importance Attacks (IFIA) proposed by [GAZ17]. We use their top-k attack with $k = 100$, adversarial budget $\epsilon = 8$, step size $\alpha = 1$ and number of iterations $P = 50$. We set the feature importance function as Integrated Gradients(IG) and dissimilarity function $D$ as Kendall's rank order correlation. We find that IFIA is not stable if we use GPU parallel computing (non-deterministic is a behavior of GPU), so we run IFIA three times on each test example and use the best result with the lowest Kendall's rank order correlation.

### C.1.4 Flower

**Data**. Flower dataset [NZ06] is a dataset of 17 category flowers with 80 images for each class (1,360 image in total). The flowers chosen are some common flowers in the UK. The images have large scale, pose and light variations and there are also classes with large variations of images within the class and close similarity to other classes. We randomly split the dataset into training and test sets. The training set has totally 1,224 images with 72 images per class. The test set has totally 136 images with 8 images per class. We resize each image to $128 \times 128$. The pixel values are in range of $[0, 255]$. We use data augmentation techniques to enlarge the training data. We construct a class preserving data augmentation pipeline consisting of rotation, translation, and projection transforms and apply this pipeline to images in the training set until each class contained 1,000 training examples. We use this new augmented training data set containing 17,000 samples in total to train models.

**Model**. We use the Resnet model [HZRS16]. We perform per image standardization before feeding images to the neural network. The network has 5 residual units with (16, 16, 32, 64) filters each. The model is adapted from CIFAR-10 model of [MMS$^+$17b]. Refer to our codes for details.

**Training hyper-parameters**. The hyper-parameters to train different models are listed below:

*NATURAL*. We use Momentum Optimizer with weight decay. We set momentum rate as 0.9, weight decay rate as 0.0002, batch size as 16, and training steps as 70,000. We use learning rate schedule: the first 500 steps, we use learning rate of $10^{-3}$; after 500 steps and before 60,000 steps, we use learning rate of $10^{-2}$; after 60,000 steps, we use learning rate of $10^{-3}$.

*Madry et al.*. We use Momentum Optimizer with weight decay. We set momentum rate as 0.9, weight decay rate as 0.0002, batch size as 16, and training steps as 70,000. We use learning rate schedule: the first 500 steps, we use learning rate of $10^{-3}$; after 500 steps and before 60,000 steps, we use learning rate of $10^{-2}$; after 60,000 steps, we use learning rate of $10^{-3}$. We use PGD attack as adversary with random start, number of steps of 7, step size of 2, and adversarial budget $\epsilon$ of 8.

*IG-NORM*. We set $\lambda$ as 0.1, $m$ in the gradient step as 50. We use Momentum Optimizer with weight decay. We set momentum rate as 0.9, weight decay rate as 0.0002, batch size as 16, and training steps as 70,000. We use learning rate schedule: the first 500 steps, we use learning rate of $10^{-4}$; after 500 steps and before 60,000 steps, we use learning rate of $10^{-3}$; after 60,000 steps, we use learning rate of $10^{-4}$. We use PGD attack as adversary with random start, number of steps of 7, step size of 2, $m$ in the attack step of 5, and adversarial budget $\epsilon$ of 8.

*IG-SUM-NORM*. We set $\beta$ as 0.1, $m$ in the gradient step as 50. We use Momentum Optimizer with weight decay. We set momentum rate as 0.9, weight decay rate as 0.0002, batch size as 16, and training steps as 70,000. We use learning rate schedule: the first 500 steps, we use learning rate of $10^{-4}$; after 500 steps and before 60,000 steps, we use learning rate of $10^{-3}$; after 60,000 steps, we use learning rate of $10^{-4}$. We use PGD attack as adversary with random start, number of steps of 7, step size of 2, $m$ in the attack step of 5, and adversarial budget $\epsilon$ of 8.

**Evaluation Attacks**. For attacking inputs to change model predictions, we use PGD attack with number of steps of 40, adversarial budget $\epsilon$ of 8 and step size of 2. For attacking inputs to change interpretations, we use Iterative Feature Importance Attacks (IFIA) proposed by [GAZ17]. We use their top-k attack with $k = 1000$, adversarial budget $\epsilon = 8$, step size $\alpha = 1$ and number of iterations $P = 100$. We set the feature importance function as Integrated Gradients(IG) and dissimilarity function $D$ as Kendall's rank order correlation. We find that IFIA is not stable if we use GPU parallel computing (non-deterministic is a behavior of GPU), so we run IFIA three times on each test example and use the best result with the lowest Kendall's rank order correlation.

## C.2 Why a different $m$ in the Attack Step?

From our experiments, we find that the most time consuming part during training is using adversary $\mathcal{A}$ to find $\boldsymbol{x}^*$. It is because we need to run several PGD steps to find $\boldsymbol{x}^*$. To speed it up, we set a smaller $m$ (no more than 10) in the attack step.

## C.3 Choosing Hyper-parameters

Our IG-NORM (or IG-SUM-NORM) objective contains hyper-parameters $m$ in the attack step, $m$ in the gradient step and $\lambda$ (or $\beta$). From our experiments, we find that if $\lambda$ (or $\beta$) is too large, the training cannot converge. And if $\lambda$ (or $\beta$) is too small, we cannot get good attribution robustness. To select best $\lambda$ (or $\beta$), we try three values: 1, 0.1, and 0.01, and use the one with the best attribution robustness. For $m$ in the attack step, due to the limitation of computing power, we usually set a small value, typically 5 or 10. We study how $m$ in the gradient step affects results on MNIST using IG-NORM objective. We try $m \in \{10, 20, 30, \cdots, 100\}$, and set $\lambda = 1$ and $m$ in the attack step as 10. Other training settings are the same. The results are summarized in Table 3.

| $m$ | NA | AA | IN | CO |
|-----|--------|--------|---------|--------|
| 10 | 98.54% | 78.05% | 67.14% | 0.2574 |
| 20 | 98.72% | 80.29% | 70.78% | 0.2699 |
| 30 | 98.70% | 80.44% | 71.06% | 0.2640 |
| 40 | 98.79% | 73.41% | 64.76% | 0.2733 |
| 50 | 98.74% | 81.43% | **71.36%** | **0.2841** |
| 60 | 98.78% | 89.25% | 63.55% | 0.2230 |
| 70 | 98.80% | 74.78% | 67.37% | 0.2556 |
| 80 | 98.75% | 80.26% | 69.90% | 0.2633 |
| 90 | 98.61% | 78.54% | 70.88% | 0.2787 |
| 100 | 98.59% | 89.36% | 59.70% | 0.2210 |

Table 3: Experiment results for different $m$ in gradient step on MNIST.

From the results, we can see when $m = 50$, we can get the best attribution robustness. For objective IG-SUM-NORM and other datasets, we do similar search for $m$ in the gradient step. We find that usually, $m = 50$ can give good attribution robustness.

## C.4 Dimensionality and effectiveness of attribution attack

Similar to [GAZ17], we observe that IFIA is not so successful when number of dimensions is relatively small. For example, on GTSRB dataset the number of dimensions is relatively small $(32 \times 32 \times 3)$, and if one uses small adversarial budget $(8/255 \approx 0.031)$, the attacks become not very effective. On the other hand, even though MNIST dimension is small $(28 \times 28 \times 1)$, the attack remains effective for large budget $(0.3)$. On Flower dataset the number of dimension is large $(128 \times 128 \times 3)$, and the attack is very effective on this dataset.

## C.5 Use Simple Gradient to Compute Feature Importance Maps

We also experiment with Simple Gradient (SG) [SVZ13] instead of Integrated Gradients (IG) to compute feature importance map. The experiment settings are the same as previous ones except that we use SG to compute feature importance map in order to compute rank correlation and top intersection, and also in the Iterative Feature Importance Attacks (IFIA) (evaluation attacks). The results are summarized in Table 4. Our method produces significantly better attribution robustness than both natural training and adversarial training, except being slightly worse than adversarial training on Fashion-MNIST. We note that Fashion-MNIST is also the only data set in our experiments where IG results are significantly different from that of SG (where under IG, IG-SUM-NORM is significantly better). Note that IG is a *princpled sommothed verison* of SG and so this result highlights differences between these two attribution methods on a particular data set. More investigation into this phenomenon seems warranted.

| Dataset | Approach | NA | AA | IN | CO |
|---|---|---|---|---|---|
| MNIST | NATURAL | 99.17% | 0.00% | 16.64% | 0.0107 |
| | Madry et al. | 98.40% | 92.47% | 47.95% | 0.2524 |
| | IG-SUM-NORM | 98.34% | 88.17% | **61.67%** | **0.2918** |
| Fashion-MNIST | NATURAL | 90.86% | 0.01% | 21.55% | 0.0734 |
| | Madry et al. | 85.73% | 73.01% | **58.37%** | **0.3947** |
| | IG-SUM-NORM | 85.44% | 70.26% | 54.91% | 0.3674 |
| GTSRB | NATURAL | 98.57% | 21.05% | 51.31% | 0.6000 |
| | Madry et al. | 97.59% | 83.24% | 70.27% | 0.6965 |
| | IG-SUM-NORM | 95.68% | 77.12% | **75.03%** | **0.7151** |
| Flower | NATURAL | 86.76% | 0.00% | 6.72% | 0.2996 |
| | Madry et al. | 83.82% | 41.91% | 54.10% | 0.7282 |
| | IG-SUM-NORM | 82.35% | 47.06% | **65.59%** | **0.7503** |

Table 4: Experiment results for using Simple Gradient to compute feature importance maps.

## C.6 Additional Visualization Results

Here we provide more visualization results for MNIST in Figure 3, for Fashion-MNIST in Figure 4, for GTSRB in Figure 5, and for Flower in Figure 6.

| NATURAL | IG-NORM | IG-SUM-NORM |
|---------|---------|-------------|
| Top-100 Intersection: 37.0% | Top-100 Intersection: 64.0% | Top-100 Intersection: 67.0% |
| Kendall's Correlation: 0.0567 | Kendall's Correlation: 0.1823 | Kendall's Correlation: 0.2180 |

(a) For all images, the models give *correct* prediction – 6.

| Top-100 Intersection: 43.0% | Top-100 Intersection: 74.0% | Top-100 Intersection: 84.0% |
|---|---|---|
| Kendall's Correlation: 0.0563 | Kendall's Correlation: 0.1718 | Kendall's Correlation: 0.2501 |

(b) For all images, the models give *correct* prediction – 3.

| Top-100 Intersection: 41.0% | Top-100 Intersection: 83.0% | Top-100 Intersection: 84.0% |
|---|---|---|
| Kendall's Correlation: 0.1065 | Kendall's Correlation: 0.2837 | Kendall's Correlation: 0.3151 |

(c) For all images, the models give *correct* prediction – 2.

Figure 3: Top-100 and Kendall's Correlation are rank correlations between original and perturbed saliency maps. NATURAL is the naturally trained model, IG-NORM and IG-SUM-NORM are models trained using our robust attribution method. We use attribution attacks described in [GAZ17] to perturb the attributions while keeping predictions intact. For all images, the models give *correct* predictions. However, the saliency maps (also called feature importance maps), computed via IG, show that attributions of the naturally trained model are very fragile, either visually or quantitatively as measured by correlation analysis, while models trained using our method are much more robust in their attributions.

NATURAL        IG-NORM        IG-SUM-NORM

Top-100 Intersection: 50.0%    Top-100 Intersection: 63.0%    Top-100 Intersection: 87.0%
Kendall's Correlation: 0.4595    Kendall's Correlation: 0.6099    Kendall's Correlation: 0.6607

(a) For all images, the models give *correct* prediction – Ankle boot.

Top-100 Intersection: 47.0%    Top-100 Intersection: 54.0%    Top-100 Intersection: 65.0%
Kendall's Correlation: 0.1293    Kendall's Correlation: 0.2508    Kendall's Correlation: 0.3136

(b) For all images, the models give *correct* prediction – Sandal.

Top-100 Intersection: 39.0%    Top-100 Intersection: 61.0%    Top-100 Intersection: 71.0%
Kendall's Correlation: 0.4129    Kendall's Correlation: 0.5983    Kendall's Correlation: 0.6699

(c) For all images, the models give *correct* prediction – Trouser.

Figure 4: Top-100 and Kendall's Correlation are rank correlations between original and perturbed saliency maps. NATURAL is the naturally trained model, IG-NORM and IG-SUM-NORM are models trained using our robust attribution method. We use attribution attacks described in [GAZ17] to perturb the attributions while keeping predictions intact. For all images, the models give *correct* predictions. However, the saliency maps (also called feature importance maps), computed via IG, show that attributions of the naturally trained model are very fragile, either visually or quantitatively as measured by correlation analysis, while models trained using our method are much more robust in their attributions.

NATURAL                    IG-NORM                    IG-SUM-NORM

Top-100 Intersection: 45.0%     Top-100 Intersection: 78.0%     Top-100 Intersection: 80.0%
Kendall's Correlation: 0.5822   Kendall's Correlation: 0.7471   Kendall's Correlation: 0.7886

(a) For all images, the models give *correct* prediction – Dangerous Curve to The Left.

Top-100 Intersection: 56.0%     Top-100 Intersection: 85.0%     Top-100 Intersection: 83.0%
Kendall's Correlation: 0.6679   Kendall's Correlation: 0.7963   Kendall's Correlation: 0.8338

(b) For all images, the models give *correct* prediction – General Caution.

Top-100 Intersection: 43.0%     Top-100 Intersection: 67.0%     Top-100 Intersection: 81.0%
Kendall's Correlation: 0.6160   Kendall's Correlation: 0.7595   Kendall's Correlation: 0.8128

(c) For all images, the models give *correct* prediction – No Entry.

Figure 5: Top-100 and Kendall's Correlation are rank correlations between original and perturbed saliency maps. NATURAL is the naturally trained model, IG-NORM and IG-SUM-NORM are models trained using our robust attribution method. We use attribution attacks described in [GAZ17] to perturb the attributions while keeping predictions intact. For all images, the models give *correct* predictions. However, the saliency maps (also called feature importance maps), computed via IG, show that attributions of the naturally trained model are very fragile, either visually or quantitatively as measured by correlation analyses, while models trained using our method are much more robust in their attributions.

NATURAL

IG-NORM

IG-SUM-NORM

Top-1000 Intersection: 1.0%
Kendall's Correlation: 0.4601

Top-1000 Intersection: 65.4%
Kendall's Correlation: 0.7248

Top-1000 Intersection: 63.9%
Kendall's Correlation: 0.8036

(a) For all images, the models give *correct* prediction – Bluebell.

Top-1000 Intersection: 6.2%
Kendall's Correlation: 0.3863

Top-1000 Intersection: 58.20%
Kendall's Correlation: 0.6694

Top-1000 Intersection: 65.9%
Kendall's Correlation: 0.7970

(b) For all images, the models give *correct* prediction – Cowslip.

Top-1000 Intersection: 6.8%
Kendall's Correlation: 0.4653

Top-1000 Intersection: 58.0%
Kendall's Correlation: 0.7165

Top-1000 Intersection: 63.4%
Kendall's Correlation: 0.8201

(c) For all images, the models give *correct* prediction – Tigerlily.

Figure 6: Top-1000 and Kendall's Correlation are rank correlations between original and perturbed saliency maps. NATURAL is the naturally trained model, IG-NORM and IG-SUM-NORM are models trained using our robust attribution method. We use attribution attacks described in [GAZ17] to perturb the attributions while keeping predictions intact. For all images, the models give *correct* predictions. However, the saliency maps (also called feature importance maps), computed via IG, show that attributions of the naturally trained model are very fragile, either visually or quantitatively as measured by correlation analyses, while models trained using our method are much more robust in their attributions.