[Reviews · NeurIPS 2019]

Reviewer 1



In this paper, the authors focus on the problem preventing attribution attacks (small input modifications which don't change the prediction but change the attribution). They do this by exploring two different framings of robust optimization that consider the Path Integrated Gradients, (P)IGs, for examples near the original example. Most interesting to me (although not an expert in this subfield), is the degree to which the paper connects robust predictions and robust attribution; showing they are equivalent in simple NNs, explaining how robust prediction optimization as in Madry et al. allows for "cancelling out" of attribution shifts, and demonstrating empirically that optimizing for robust attribution improves robust predictions. I'd encourage the authors to draw this out more. The biggest weakness for me is does this approach change my interpretation of attributions? Following the example from the intro about pathology, should I trust now the attributions from a model that is designed to not be sensitive to small perturbations? Attacking attributions demonstrates their brittleness and shows that interpreting the "reasoning" of a NN from these attributions should be taken with a grain of salt, but it is unclear to me if this is a surface-level fix to a deeper rooted problem with attribution or if this is directly addressing a central problem. So while the theoretical connection is interesting, I'd like to see a clearer argument for why designing models that have robust attribution is an important property. Overall, this paper is interesting and in my opinion a valuable contribution. Details: Sec 3.2 is gratuitous to me. The objective is never used and at least as written I didn't find the connection as intriguing. Is there any functional value to the use of an intermediate lay r for IG in (4)? Table 2 is redundant with Table 3.

Reviewer 2



I have scored the paper a 7 since I feel that this is a well-written paper on an interesting and relevant topic. Kudos to the authors for clear exposition, including code, and providing some guidance on how to set regularization parameters. While I don't have much to critique, I do have two comments: 1. I do think that the paper could be *much* stronger if the authors could present some negative results regarding "robust" saliency maps. For example, it is possible to use their methods to build saliency maps that are robust to perturbations, yet are completely flawed in their attributions? This would make a nice use case for the method - giving us a sanity check to prevent rationalization. In line with this comment, it is worth adding a short note on the issues that are not fixed in with saliency maps (see e.g., https://arxiv.org/abs/1901.09749, https://arxiv.org/abs/1811.10154). 2. In some sense the previous comment highlights the following issue, which I would mention in the text. This method "appears to work" since the experiments report metrics that are bound to improve given the constraints (e.g., Kendall’s correlation between a saliency map and perturbed saliency maps is bound to improve given the set of constraints). While I don't think that this is a limitation of the work, I think that the paper should mention this explicitly for the sake of transparency. In short, it would help new readers to know that we are only fixing a form of brittleness that we can measure, and that it will not necessarily fix the more difficult attribution problem (though it can screen away clear cases of misattribution). Other issues: - l.183 One-Layer <- Single-Layer) - p.4 footnote: "We stress that this regularization term depends on model parameters θ through loss function l[y]" <- what does this mean?

Reviewer 3



The work studies an important problem, that of optimizing networks with respect to attribution robustness. The problem, in a way, is not only related to interpretability for its own sake, but also for improving the robustness of models in general, and that connection is very well phrased in the paper. Comments: - The paper would benefit by getting another, maybe richer, dataset in the evaluation. MNIST is not a great example, especially when it comes to interpretations and robustness. - In general, it is unclear (even from previous work in this space), how does attribution robustness correlate to the human perception of model interpretability. I am not aware of studies that have tried to measure this (empirically) but if there are, it is very useful including them in the paper. If such studies do not exist, it would be beneficial to at least have a paragraph that analytically explains how close this may or may not be to human interpretation. It is fair for the reader to know whether implementing such optimizations in practice, would even be visible to people at all. On the same point, it is hard to understand how the improvements in IN and CC (Table 3) relate to practical improvements in robustness. Does an improvement of 0.02 really make the model outputs more interpretable to the input changes? - On the ineffective optimization paragraph, point (2). This point deserves further and more precise discussion on why the authors think that the architecture is not suitable. Also, it needs a clarification on whether "architecture" here means the generic nature of NNs or the particular architecture of the networks studied in this paper. - Minor: You could actually write the actual name of the metric in the header of Table 3 instead of the Accronyms.

[Author Response · NeurIPS 2019]

We thank the reviewers for encouraging and insightful comments. Below we respond to reviewers' major comments.

**Question 1 (R1, R4, R5)** *Why study robust attribution regularization? How does it correlate with human perception? (e.g., how about robust but totally flawed attributions?)*

We appreciate this question about the essential motivation of this work. To begin with, we believe that robust attribution is at least *necessary* for a machine learning model to be trustworthy. Model attributions are *facts* about a model's behaviors. It is true, as **R1** pointed out, that users may still want to be skeptical when interpreting attributions, even if they are robust. However in the opposite direction, one can never trust a model with *brittle* attributions.

Regarding human perception, a very intriguing recent paper [3] from Madry's group showed that, empirically, adversarial training (or robust prediction training) produces models whose attributions are *much more aligned with human perception*, and seems to learn salient features from data. Since robust prediction training is a special case of robust attribution training, their results form a basis for our belief that robust attribution regularization can lead to better correlation with human perception (which is however a somewhat subjective question). Indeed, our visualization results in the draft and appendices also corroborate their findings.

Note that, in the worst case, it is easy to construct models that have robust but totally flawed attributions – simply take models that always have "the same" behaviors. However, the situation becomes much more complex if one imposes the constraint that the model should also achieve small training error, which is what we did in this paper. Therefore, robust attributions can be thought of as imposing an inductive bias to encourage learning *invariant* features from data. Since human perception is essentially also related to recognizing invariances, small training error with invariant features intuitively implies alignment with human perception. Next version will incorporate the above points.

**Question 2 (R4, R5)** *How useful is evaluating IN and CC metrics?*

The use of IN and CC is aligned with previous literature in studying robustness of attributions (in particular, the work of Ghorbani et al. [1]). We agree with reviewers that these two metrics are only one form of brittleness, and are direct consequences of our objectives. We also agree with reviewers that it is hard to quantify how increasing these two metrics improves (directly) human perception. On the other hand, we think that these evaluations are still useful: (1) It answers the scientific questions raised in [1], (2) Perhaps more importantly, it corroborates our analysis that robust prediction training will robustify attributions as well. We will make these points explicit in the next version.

**Question 3 (R1)** *Section 3.2 seems redundant.*

We apologize for the confusion. In fact Section 3.2 fulfills an important theoretical purpose: Distributional robustness approach forms a different school towards robust prediction training (see [2]). The analysis here shows that generalizing these objectives to robust attributions (a much larger class) essentially still gives very similar objectives in two different robust optimization models, and thus we can "safely" stick to the formulation in Section 3.1, which is reassuring.

**Question 4 (R1)** *Is there any functional value of regularizing an intermediate layer?*

Proposition 3 proves that if one regularizes by the output layer it gives a natural surrogate loss of Madry et al.'s objective function, which to us makes even more sense as it directly bounds the *absolute difference* between $x$ and $x'$. We believe that there is more to regularizing intermediate layers and we are actively researching it.

**Question 5 (R5)** *More datasets, MNIST is not great.*

Besides MNIST we have evaluated Flower and GTSRB (traffic signs). Both are more diverse than MNIST. GTSRB is practically motivated, and Flower is a high-resolution vision dataset well suited for studying attributions. We get similar results in terms of both metrics and visualizations. We are actively working on more datasets and plan to include more in the final version if this paper gets in.

**Question 6 (R5)** *More details on optimization difficulty and architectural properties.*

We will add more details. Roughly speaking, a main problem is *network depth*, where as depth increases we get very *unstable* trajectories of gradient descent, which seems to be related to the use of second order information during robust attribution optimization (due to summation approximation, we have first order terms in the training objectives).

# References

[1] A. Ghorbani, A. Abid, and J. Y. Zou. Interpretation of neural networks is fragile. In *The Thirty-Third AAAI Conference on Artificial Intelligence, AAAI 2019*, pages 3681–3688, 2019.

[2] A. Sinha, H. Namkoong, and J. C. Duchi. Certifying some distributional robustness with principled adversarial training. In *6th International Conference on Learning Representations, ICLR 2018*, 2018.

[3] D. Tsipras, S. Santurkar, L. Engstrom, A. Turner, and A. Madry. Robustness may be at odds with accuracy. In *7th International Conference on Learning Representations, ICLR 2019*, 2019.


[Meta-Review · NeurIPS 2019]

Overall, the reviewers found that the paper makes an intriguing and useful connection between attribution methods traditionally used for model explanability and robust predictions for improved generalization. They also found the paper to be clearly written. I did find the suggestions from R4 to search for any "negative results" around attribution brittleness to be an important one, and agree it would strengthen the paper if such examples could be found to help the reader understand more about the behavior of robust saliency maps. And I do expect that the authors will take advantage of the extra space for the camera ready version to include additional datasets and empirical results, as they described they would in the author response.